# Anti-α-Glucosidase and Antiglycation Activities of α-Mangostin and New Xanthenone Derivatives: Enzymatic Kinetics and Mechanistic Insights through In Vitro Studies

**DOI:** 10.3390/molecules27020547

**Published:** 2022-01-15

**Authors:** Francine Medjiofack Djeujo, Valeria Francesconi, Maddalena Gonella, Eugenio Ragazzi, Michele Tonelli, Guglielmina Froldi

**Affiliations:** 1Department of Pharmaceutical and Pharmacological Sciences, University of Padova, 35131 Padova, Italy; francine.medjiofackdjeujo@phd.unipd.it (F.M.D.); maddalenagonella96@gmail.com (M.G.); eugenio.ragazzi@unipd.it (E.R.); 2Department of Pharmacy, University of Genova, 16132 Genova, Italy; francesconi.phd@difar.unige.it

**Keywords:** antidiabetic agents, α-glucosidase inhibitors, glycation inhibition, natural compounds, *Garcinia mangostana*, ORAC assay, BSA assay

## Abstract

Diabetes mellitus is characterized by chronic hyperglycemia that promotes ROS formation, causing severe oxidative stress. Furthermore, prolonged hyperglycemia leads to glycation reactions with formation of AGEs that contribute to a chronic inflammatory state. This research aims to evaluate the inhibitory activity of α-mangostin and four synthetic xanthenone derivatives against glycation and oxidative processes and on α-glucosidase, an intestinal hydrolase that catalyzes the cleavage of oligosaccharides into glucose molecules, promoting the postprandial glycemic peak. Antiglycation activity was evaluated using the BSA assay, while antioxidant capacity was detected with the ORAC assay. The inhibition of α-glucosidase activity was studied with multispectroscopic methods along with inhibitory kinetic analysis. α-Mangostin and synthetic compounds at 25 µM reduced the production of AGEs, whereas the α-glucosidase activity was inhibited only by the natural compound. α-Mangostin decreased enzymatic activity in a concentration-dependent manner in the micromolar range by a reversible mixed-type antagonism. Circular dichroism revealed a rearrangement of the secondary structure of α-glucosidase with an increase in the contents of α-helix and random coils and a decrease in β-sheet and β-turn components. The data highlighted the anti-α-glucosidase activity of α-mangostin together with its protective effects on protein glycation and oxidation damage.

## 1. Introduction

The main causes of morbidity in diabetes mellitus (DM) are chronic complications resulting from prolonged hyperglycemia [1]. These can be alleviated through careful and continuous blood glucose control. Micro- and macrovascular damage occurs at the arterial level, causing cardiovascular and cerebrovascular diseases that can result in cardiac infarction and stroke [2,3]. In the condition of chronic hyperglycemia, glucose tends to form covalent adducts with plasma proteins (albumin, fibrinogen, globulins, collagen) through a nonenzymatic process known as glycation [1,4]. Protein glycation and formation of advanced glycation end products (AGEs) play a crucial role in the pathogenesis of DM-related complications such as retinopathy, nephropathy, and cardiomyopathy [1]. Recent studies suggest that AGEs interact with the plasma membrane by specific receptors (RAGE), changing intracellular signal transmission, modifying gene expression, and promoting the release of pro-inflammatory molecules and free radicals [5]. The enzyme α-glucosidase is an intestinal hydrolase located in the brush-rim membrane of intestinal cells that catalyzes the cleavage of food oligosaccharides into glucose molecules, promoting the postprandial glycemic peak. Inhibition of the enzyme activity determines a significant delay in intestinal glucose absorption by decreasing the postprandial glycemic peak and helping glycaemia control [6], also in the multi-drug combination therapy [7,8].

α-Mangostin is a 9*H*-xanthene-9-one derivative mainly isolated from the fruit of *Garcinia mangostana* L. (Clusiaceae family), commonly known as mangosteen, a plant widely cultivated in Southeast Asian countries [8,9,10]. Studies have shown that the plant-derived α-mangostin possesses several pharmacological properties, including anti-inflammatory, antidiabetic, cardioprotective, and antimicrobial activities [10,11,12,13,14]. α-Mangostin and related xanthones are generally considered safe and well tolerated in vivo [15,16].

In general, in the literature, xanthenone compounds (dibenzo-γ-pyrones) have been shown to possess numerous biological properties such as antiproliferative, anti-HIV, anticholinesterase, and antimalarial effects but are also known to have inhibitory activity against α-glucosidase [17]. Molecular modeling studies have shown that xanthenone derivatives are capable of binding to allosteric sites of α-glucosidase, mainly establishing π-stacking interactions, and when additional aromatic moieties are introduced as decorations of the main core, biological performance increases due to hydrophobic contacts with the target enzyme. Furthermore, improvements in inhibitory activity could be observed by including oxygen or nitrogen-containing groups on these aromatic units [17,18]. Applying a molecular simplification strategy to α-mangostin structure, we synthesized a small set of xanthenone derivatives (CS1–CS4), substituted at position 1 with various side chains, bearing polar groups, and with a methyl group at position 4 (Figure 1).

The purpose of this investigation was to explore the potential of the plant-derived α-mangostin and related synthetic xanthenone derivatives CS1–CS4 as antidiabetic agents. The study was performed by evaluating the following: (i) antiglycation activity using the bovine serum albumin (BSA) assay, (ii) antioxidant activity with oxygen radical absorbance capacity (ORAC) test, and (iii) anti-α-glucosidase activity. The inhibition of α-glucosidase activity was deepened through enzymatic kinetics and fluorescence studies, which have allowed us to understand the type of interaction and the inhibition mechanism.

## 2. Results and Discussion

### 2.1. α-Mangostin and Synthetic Compounds CS1–CS4: Chemistry

The ability of α-mangostin to exert antidiabetic effects was studied together with four synthetic compounds (CS1–CS4) to investigate new chemical characteristics in the xanthenone core (Figure 1).

For the synthesis of compounds CS1-CS4 a mixture of 1-chloro-4-methyl-9*H*-xanthen-9-one, prepared according to [19], with the appropriate amine derivative was melted at 170 °C in a sealed tube for 4 h (Figure 1).

The structure of the synthetic compounds was confirmed using ^1^H and ^13^C NMR and elemental analysis. The purity of the compounds checked by elemental analysis was ≥95%.

### 2.2. α-Mangostin and Synthetic Compounds CS1–CS4: Antiglycation Activity

α-Mangostin and synthetic compounds were evaluated with the BSA assay, which measures the fluorescence produced by the AGE formation from ribose and albumin reaction [20]. Aminoguanidine (25 mM) was used as a reference inhibitor (positive control) [21,22]. In this setup, glycation increased significantly after two days of incubation, while maximum AGE production was detected after 5 days, and then it became stable at 7 and 9 days (Appendix A).

Figure 2 shows the effect of α-mangostin (5–75 μM) on ribose-induced albumin glycation. The natural compound showed a concentration-dependent inhibition, with a maximum effect at 75 μM after 2 days of incubation (−30%). Inhibition was maintained at 5 and 7 days, even if it gradually decreased afterward. These results agree with a previous study showing the anti-AGE effect of some xanthones isolated from a methanol extract of *Garcinia mangostana* where 100 µM α-mangostin significantly inhibited three-day ribose-induced glycation [23].

Figure 3 reports the inhibitory activity of the synthetic xanthenones CS1–CS4. Interestingly, compounds CS2 and CS3 at the highest concentration tested produced an antiglycation effect of 32% and 25%, respectively, on day 2, which gradually decreased during incubation, showing an inhibition comparable to α-mangostin. Thus, the 1-(3-hydroxypropyl)amino and 1-(2-morpholinoethyl)amino chains of CS2 and CS3 have emerged as novel fragments of the xanthone nucleus, capable of promoting inhibition of AGE production. Otherwise, CS1 and CS4 only slightly reduced glycation at the highest concentration tested (days 2 and 5). Overall, α-mangostin and, among synthetic compounds, mainly CS2 and CS3 showed significant antiglycation activities. Previously, other authors suggested that a *G. mangostana* total methanol extract and four isolated phenolic compounds reduce fructosamine (Amadori product) and protein aggregation formation in glucose and ribose-induced BSA glycation [24]. However, the specific mechanisms involved in the antiglycation activity of the considered compounds merit future investigation.

### 2.3. α-Mangostin and Synthetic Compounds CS1–CS4: Antioxidant Activity

Since the antiglycation activity of α-mangostin and synthetic compounds could be related to their antioxidant capacity, the oxygen radical absorbance capacity (ORAC) assay was performed. This test evaluates the ability of compounds to inhibit the oxidative degradation of a fluorescent probe caused by peroxyl radicals (ROO^•^) through the hydrogen atom transfer (HAT) mechanism [25,26]. In this investigation, ascorbic acid was used as a reference inhibitor (positive control) [27].

The antiradical activity of α-mangostin was approximately four times higher than that of ascorbic acid: 2.73 ± 0.26 μmol TE/μmol and 0.69 ± 0.03 μmol TE/μmol, respectively (Figure 4). Among the synthetic compounds, CS3 showed the highest activity (2.44 ± 0.16 μmol TE/μmol), followed by CS1 (1.83 ± 0.28 μmol TE/μmol), CS2 (1.36 ± 0.31 μmol TE/μmol), and CS4 (0.24 ± 0.05 μmol TE/μmol). The order of antioxidant activity was α-mangostin > CS3 > CS1 ≥ CS2 > ascorbic acid > CS4. α-Mangostin is a natural polyphenol that has been reported to have antioxidant activity thanks to the hydroxyl groups; in particular, the authors indicated that the C-6 OH group is fundamental in HAT mechanism [28,29]. Analogous considerations have been reported for other xanthone derivatives, whose antioxidant activity was found to be affected by the number and position of hydroxyl groups [30]. In particular, although morpholinoethylamino xanthenone CS3 is devoid of any hydroxyl group, it ranks second in potency. Therefore, the exploration of novel biologically relevant regions of chemical space around the xanthone scaffold may deserve further investigation. It should be noted that antiradical activity can be useful in trapping intermediate dicarbonyl compounds, decreasing glycation and AGE formation [31].

### 2.4. Yeast α-Glucosidase Inhibitory Activity

#### 2.4.1. α-Mangostin and Synthetic Compounds CS1–CS4

The inhibitory activity of α-mangostin and synthetic compounds was studied by a colorimetric assay using pNPG as a substrate. α-Glucosidase hydrolyzes pNPG into α-D-glucopyranoside and p-nitrophenol (yellow) whose chromatic intensity was detected: staining decreases proportionally to the ability of an inhibitor to counteract enzyme activity [32].

Figure 5 A shows the absorbance curves during the 60 min reaction between 0.04 µM α-glucosidase and 2 mM pNPG in the presence of α-mangostin (1–50 µM). The compound inhibited the degradation of pNPG in a concentration-dependent manner, reducing the enzyme activity. Inhibition with 5 µM α-mangostin was 25.5 ± 2.6% (*p* < 0.001), while with 50 µM, 60.8 ± 2.5% (*p* < 0.001).

Furthermore, the inhibition of α-mangostin was assessed using 0.05 and 0.06 µM α-glucosidase. As expected, with higher enzyme concentrations, the absorbance curves moved to the left, and the plateau was reached in a shorter time. Table 1 reports the K_m_ and V_max_ values obtained with 0.05 µM α-glucosidase alone and in the presence of α-mangostin. K_m_ increased, while the maximum reaction velocity (V_m_) decreased with increasing inhibitor concentrations (Table 1), suggesting a mixed-type interaction between α-mangostin and the enzyme.

Unlike the natural compound, the compounds CS1 and CS4 tested from 1 μM to 100 μM did not inhibit the α-glucosidase activity; additionally, CS2 and CS3, which were used at lower concentrations (up to 25 μM) due to their minor solubility, were inactive (Appendix A). On this basis, it can be deduced that inhibition of α-glucosidase activity is mainly driven by the chemical nature and disposition of substituents on the xanthenone scaffold rather than the xanthenone core itself. Therefore, further investigations into α-glucosidase inhibition were only performed with α-mangostin.

To study the type of interaction between the enzyme and α-mangostin, the graph “*v*
*versus* [α-glucosidase]” was evaluated (Figure 5B), suggesting a concentration-dependent reversible interaction since all straight lines obtained at different concentrations of α-mangostin pass through the origin of the axes, and their slope decreased with increasing concentrations of the inhibitor. With 0.05 μM α-glucosidase and 2 mM pNPG, the IC_50_ of α-mangostin was 31.1 μM (pIC_50_ = 4.51 ± 0.04), showing a higher affinity than acarbose, IC_50_ = 1712 μM (pIC_50_ = 2.77±0.04), tested as a positive control, resulting in an inhibitory potency ratio of 55. Other authors reported the α-mangostin inhibition on α-glucosidase, showing IC_50_ values ranging from 1.3 to 29.3 μM [11,12,13,33,34]; among these investigations, only three compared the inhibition with that of acarbose, evidencing a potency ratio of 2.1, 8.2 and 18.8 [11,13,34].

#### 2.4.2. α-Mangostin Inhibition with Different Concentrations of pNPG

An in-depth study of the α-mangostin interaction was performed using five different concentrations of substrate pNPG, from 0.25 mM to 2 mM, while the concentration of α-glucosidase was maintained constant (0.05 µM). Figure 6 reports the graphical elaborations of Michaelis–Menten (A) and Lineweaver–Burk (B). In particular, the Lineweaver–Burk plot suggests a mixed-type inhibition, since the lines intercept in the second quadrant (Figure B). The mixed-type inhibitor has different affinities for the free enzyme and enzyme–substrate complex. Furthermore, the secondary slope plots were linear, suggesting that α-mangostin acts as a complete inhibitor [35]. Secondary plots of the slope and intercept (Figure 6C,D) of the data depicted in Figure 6 B provide a means to determine the kinetic constants describing the enzyme kinetics (Table 2). For α-mangostin, the inhibition constant of the enzyme (K_i_) and the inhibition constant of the enzyme–substrate complex (K_i′_) were, respectively, 30.33 µM and 45.60 µM, suggesting that the inhibitor preferentially binds the free enzyme, as K_i_ was lower than K_i’_. The catalytic efficiency of α-glucosidase expressed as K_cat_/K_m_ was also calculated, showing that the values decreased in the presence of α-mangostin, suggesting a decrease in the catalytic efficiency of α-glucosidase (Table 2). In general, the trend of the kinetic parameters (K_m_, K_cat_, K_i_, K_i’_, V_max_) and Figure 6 reveal a mixed-type inhibition of α-mangostin on α-glucosidase.

#### 2.4.3. α-Glucosidase Fluorescence Quenching by α-Mangostin

Furthermore, for the first time, the effects of α-mangostin on the secondary structure of the enzyme have been investigated using circular dichroism and synchronous fluorescence spectroscopic techniques. α-Glucosidase showed an emission peak at 340 nm, as previously reported [36,37]. Figure 7 shows the high fluorescence intensity of α-glucosidase at 344 nm when excited at 280 nm, detected at 298, 304, and 310 K, alone and in the presence of α-mangostin. It can be observed that the fluorescence intensity progressively decreased as the concentration of α-mangostin increased, suggesting a specific interaction. Stern–Volmer constant (K_sv_) and bimolecular quenching constant (K_q_) were determined using the graphs F_0_/*F* vs. [Q], Figure 7 D. It is known that fluorophore quenching can occur by dynamic and/or static quenching processes [38]. These types of mechanism can be distinguished by their different dependences on temperature and excited-state lifetime. With α-mangostin, the K_sv_ and K_q_ values are positively correlated with temperature, and the K_q_ values were three orders of magnitude higher than the limiting diffusion constant K_dif_ of the biomolecule (K_dif_ = 2.0 × 10^10^ M^−1^ s^−1^) [38], confirming a specific interaction between α-mangostin and the enzyme (Table 3) [39]. Furthermore, these results indicate that the quenching of α-glucosidase by α-mangostin is a static–dynamic process driven mainly by hydrophobic interactions, as previously shown for other compounds, such as procyanidins [40]. No other studies have been found in the literature about α-mangostin fluorescence quenching.

The possibility of obtaining characteristic information on the tyrosine and tryptophan residues of α-glucosidase using synchronous fluorescence spectroscopy has been described [41]. In detail, yeast α-glucosidase contains 589 amino acids with 20 tryptophan and 27 tyrosine residues [37]. Figure 8 shows the synchronous fluorescence spectra of the tyrosine (A) and tryptophan (B) residues with different concentrations of α-mangostin (0.05–1.5 µM). The shift from 285 to 287.5 nm for α-mangostin (Figure 8A) was observed with ∆λ = 15 nm, while there was no modification with ∆λ = 60 nm (Figure 8B), suggesting that the enzyme–ligand interaction did not influence the microenvironment of tryptophan residues. Therefore, the results show a change in the microenvironment of tyrosine, indicating that α-mangostin affected the structure of α-glucosidase.

#### 2.4.4. Circular Dichroism Measurements

Circular dichroism (CD) spectroscopy was used for the characterization of the secondary structure of α-glucosidase and to estimate its changes during interaction with α-mangostin. According to previous studies [42,43], enzyme CD spectra reported two negative minimum values at 209 and 222 nm, which coincide with α-helix structure (Figure 9). The presence of α-mangostin (molar ratio 1:1 and 1:2) increased both negative humped peaks compared to α-glucosidase alone, indicating that the α-helix content increases, while β-sheet and β-turn decrease (Table 4). No published dichroism studies have been found for α-mangostin. The increase in α-helix and random coil contents suggests that the enzyme structure tends to be more compact in the presence of α-mangostin, with a decrease in the catalytic activity of α-glucosidase, as shown, e.g., for oleanolic and ursolic acids [44].

## 3. Materials and Methods

### 3.1. Reagents

Acarbose, α-glucosidase (EC 3.2.1.20, *Saccharomyces cerevisiae* type I, 10 U/mg protein), *p*-nitrophenyl-α-d-glucopyranoside (pNPG), α-mangostin (1,3,6-trihydroxy-7-methoxy-2,8-bis(3-methylbut-2-en-1-yl)-9*H*-xanthen-9-one). All chemicals and solvents were purchased from Merck KGaA, Darmstadt, Germany. The purity of the reference standards was ≥97%, while other chemicals were of at least of analytical grade.

### 3.2. Chemistry

#### 3.2.1. General Information

Melting points (uncorrected) were determined using a Büchi apparatus (Milan, Italy). ^1^H NMR spectra and ^13^C NMR spectra were recorded by a Jeol instrument (Milan, Italy) at 400 and 101 MHz, respectively; chemical shifts were reported as δ (ppm) and referenced to the solvent signal: DMSO-*d*_6_, quintet at 2.5 ppm (^1^H), septet at 39.5 ppm (^13^C); *J* in Hz. Elemental analyses were performed on a Flash 2000 CHNS instrument (Thermo Scientific, Milan, Italy) in the Microanalysis Laboratory of the Department of Pharmacy, University of Genova. The results of the elemental analyses indicated that the purity of all compounds was >95%.

#### 3.2.2. General Synthesis of Compounds CS1–CS4

A mixture of 1-chloro-4-methyl-9*H*-xanthen-9-one (1.3 mmol) with the proper amino compound (2.6 mmol) was heated at 170 °C in a sealed tube for 4 h. After cooling, the mixture was treated with 2 N HCl and extracted with CH_2_Cl_2_ (20 mL × 3). The acid solution was then alkalinized with 6 N NaOH and extracted exhaustively extracted with CH_2_Cl_2_ (20 mL × 4). The organic layer was dried with anhydrous Na_2_SO_4_, filtered, and evaporated under vacuum, producing a yellow residue that was purified by CC (SiO_2_/CH_2_Cl_2_ + 5%MeOH). While compounds CS2–CS4 were obtained as yellow-orange crystalline solids, CS1 was an orange oil, which was converted into its corresponding monohydrochloride salt by the addition of an equimolar amount of 1 N ethanolic solution of HCl.

*1-{[4-(2-Hydroxyethyl)piperazin-1-yl]amino}-4-methyl-9H-xanthen-9-one hydrochloride* (CS1), Yield: 28%; m.p. 194–195 °C (monohydrochloride salt); anal calculation for C_20_H_23_N_3_O_3_·HCl: % C 61.61; H 6.20; N 10.78; found % C 61.81; H 5.94; N 10.11. ^1^H NMR (400 MHz, DMSO-*d*_6_): δ 10.37 (s, 1H, NH), 8.05 (dd, *J* = 7.9, 1.6 Hz, 1H, H(8)), 7.62–7.74 (ddd, *J* = 8.6, 7.1, 1.7 Hz, 1H, H(6)), 7.54 (dd, *J* = 8.3 Hz, 2H, H(3,5)), 7.36 (t, *J* = 7.5 Hz, 1H, H(7)), 6.83 (d, *J* = 8.2 Hz, 1H. H(2)), 3.78 (pseudo s, superimposed on the DMSO signal, 2H, CH_2_-OH), 3.58 (d, *J* = 11.5 Hz, 2H, N-CH_2_-CH_2_OH), 3.38 (d, *J* = 13.2 Hz, 4H, NH-N(CH_2_)_2_ piperazine), 3.26 (t, *J* = 5.2 Hz, 2H, N-(CH_2_)_2_ H_α_ piperazine), 3.21–3.11 (m, 2H, N-(CH_2_)_2_ H_β_ piperazine), 2.46 (t, *J* = 1.8 Hz, 1H, OH), 2.35 (s, 3H, CH_3_). ^13^C NMR (101 MHz, DMSO-*d*_6_) δ 208.76, 173.53, 157.86, 154.58, 150.50, 136.25, 134.82, 127.14, 124.29, 122.42, 120.33, 116.69, 114.28, 113.08, 58.98, 56.36, 52.40, 48.45, 31.21, 18.29.

*1-[(3- Hydroxypropyl)amino]-4-methyl-9H-xanthen-9-one* (CS2), Yield: 64%; m.p. 145–146 °C; Anal calculation for C_16_H_15_NO_3_: % C 72.07; H 6.05; N 4.94; found % C 72.32; H 6.05; N 4.77. ^1^H NMR (400 MHz, CDCl_3_): δ 9.33 (s, 1H, NH), 8.19 (dd, *J* = 7.9, 1.7 Hz, 1H, H(8)), 7.62 (p, *J* = 7.8, 1.6 Hz, 1H, H(6)), 7.39 (d, *J* = 8.3 Hz, 1H, H(3)), 7.33–7.23 (m, 2H, H(7,5)), 6.33 (d, *J* = 8.5 Hz, 1H, H(2)), 3.85 (t, *J* = 6.1 Hz, 2H, NH-CH_2_-), 3.35 (q, *J* = 6.4 Hz, 2H, -CH_2_-OH), 2.31 (s, 3H, CH_3_), 1.99 (p, *J* = 6.4 Hz, 2H, -CH_2_-CH_2_-CH_2_-), 1.78 (s, 1H, OH). ^13^C NMR (101 MHz, CDCl_3_): δ 180.30, 155.54, 155.44, 150.35, 137.68, 134.10, 126.05, 123.53, 122.04, 117.39, 110.43, 107.07, 103.15, 60.67, 39.82, 31.75, 15.12.

*4-Methyl-1-((2-morpholinoethyl)amino)-9H-xanthen-9-one* (CS3), Yield: 40%; m.p. 125–126 °C; Anal calculation for C_16_H_15_NO_3_: % C 70.99; H 6.55; N 8.28; found % C 70.68; H 5.99; N 7.97. ^1^H NMR (400 MHz, CDCl_3_): δ 9.42 (s, 1H, NH), 8.22 (dd, *J* = 8.0, 1.8 Hz, 1H, H(8)), 7.66–7.56 (m, 1H, H(6)), 7.38 (d, *J* = 8.4 Hz, 1H, H(3)), 7.32–7.25 (m, 2H, H(7,5)), 6.31 (d, *J* = 8.5 Hz, 1H, H(2)), 3.77 (t, *J* = 4.7 Hz, 4H, -(CH_2_)_2_O morpholine), 3.34 (q, *J* = 6.6 Hz, 2H, NH-CH_2_CH_2_- morpholine), 2.72 (t, *J* = 6.6 Hz, 1H, NH-CH_2_CH_2_- morpholine), 2.55 (pseudo s, 4H, -(CH_2_)_2_N morpholine), 2.32 (s, 3H CH_3_). ^13^C NMR (101 MHz, CDCl_3_): δ 180.13, 155.60, 155.43, 150.07, 137.54, 134.06, 126.18, 123.49, 122.13, 117.38, 110.63, 107.25, 103.14, 67.07 (2C), 57.13, 53.71 (2C), 40.11, 15.15.

*1-((Benzo[d][1,3]dioxol-5-ylmethyl)amino)-4-methyl-9H-xanthen-9-one* (CS4), Yield: 30%; m.p. 154–156 °C; Anal calculation for C_16_H_15_NO_3_: % C 75.53; H 4.77; N 3.90; found % C 75.20; H 4.50; N 3.64. ^1^H NMR (400 MHz, CDCl_3_): δ 9.70 (s, 1H, NH), 8.21 (dd, *J* = 8.0, 1.7 Hz, 1H, H(8)), 7.64 (ddd, *J* = 8.6, 7.1, 1.7 Hz, 1H, H(6)), 7.41 (d, *J* = 8.3 Hz, 1H, H(3)), 7.33–7.28 (m, 1H, H(7)), 7.27–7.23 (m, 1H, H(5)), 6.90–6.82 (m, 2H, H(2) and H(4′) benzodioxol), 6.76 (d, *J* = 7.9 Hz, 1H, H(6′) benzodioxol), 6.28 (d, *J* = 8.4 Hz, 1H, H(7′) benzodioxol), 5.92 (s, 2H, O-CH_2_-O), 4.37 (d, *J* = 5.6 Hz, 2H, NH-CH_2_- benzodioxol), 2.32 (s, 3H, CH_3_). ^13^C NMR (101 MHz, CDCl_3_): δ 180.39, 155.54, 155.48, 149.93, 148.06, 146.79, 137.58, 134.19, 132.65, 126.12, 123.58, 122.07, 120.40, 117.45, 111.05, 108.46, 107.87, 107.32, 103.74, 101.08, 47.04, 15.16, 4.5.

### 3.3. Detection of Advanced Glycation End Products

The preparation of glycated serum albumin was performed according to a previously described method [45,46]. Shortly, AGEs were determined using BSA (50 mg/mL, pH 7.4) as protein substrate and ribose (0.1 M) as glycation agent. Each compound tested from 5 to 75 μM was added in BSA-ribose reaction and incubated at 37 °C for 2, 5, 7, and 9 days. The fluorescence intensity was measured at the excitation wavelength of 355 nm and emission wavelength of 460 nm with a PerkinElmer Victor Nivo microplate reader (Waltham, MA, USA). Aminoguanidine (25 mM) was used as a positive control. The inhibition of AGE formation was calculated as the fluorescence difference between glycation under the control condition and in the presence of the inhibitor, expressed as a percentage.

### 3.4. Measurement of the Oxygen Radical Absorbance Capacity

The oxygen radical absorbance capacity (ORAC) assay allows evaluating the ability of substances to interfere with oxidative reactions induced by peroxidic radicals. The assay was performed as previously reported [47,48]. Briefly, 6-hydroxy-2,5,7,8-tetramethylchroman-2-carboxylic acid (trolox) was prepared in phosphate buffer in a concentration range of 6.25–50 μM. Fluorescein (1.5 mL of 0.08 μM solution) was added to 24-well plates, followed by 250 μL of trolox for the controls, 250 μL of buffer for the blank, and 250 μL for the samples (5.0 and 10 μg/mL). After 10 min of incubation at 37 °C, 250 μL of 0.15 M 2,2′-azobis(2-amidinopropane)-dihydrochloride (AAPH) were added. Successively, the PerkinElmer Victor Nivo microplate reader (Waltham, MA, USA) was settled for a fluorescence kinetic reading at 37 °C for 45 min, with excitation and emission wavelengths of 485 and 530 nm, respectively. Data were expressed as TEAC (trolox equivalent antioxidant capacity, TE μmol/ μmol compound).

### 3.5. Measurement of α-Glucosidase Activity

The assay was used to detect the ability of each compound (5–100 µM) to reduce the activity of yeast α-glucosidase (EC 3.2.1.20, *Saccharomyces cerevisiae* type I, 10 U/mg protein) in the presence of the substrate *p*-nitrophenyl-α-d-glucopyranoside (pNPG) [49]. Acarbose (1.25 M) was used as a positive control. Each sample was incubated with α-glucosidase 0.4, 0.5, and 0.6 µM and 0.1 M PBS (pH 6.8) for 10 min, at 37 °C. The reaction started by adding pNPG. Absorbance values were detected at 405 nm for 45 min, using a PerkinElmer Victor Nivo microplate reader (Waltham, MA, USA). The α-glucosidase activity in the absence of inhibitors was defined as 100%. Half-maximal inhibitory concentration (IC_50_) was estimated by plot of relative enzyme activity *versus* inhibitor concentration. The type of enzyme inhibition exerted by α-mangostin was evaluated from kinetic studies using different substrate concentrations (0.25–2.0 mM pNPG) applying Michaelis–Menten and Lineweaver–Burk plots.

#### 3.5.1. Fluorescence Quenching Analysis

The interaction of α-mangostin with α-glucosidase was studied using the fluorescence quenching method [50]. The fluorescence of α-glucosidase alone and in the presence of the inhibitor was studied at different concentrations (Jasco FP-6500 spectrofluorometer, Japan). Measurements were made in the emission range of 300–500 nm, with an excitation of 280 nm, after 10 min of stabilization. The fluorescent spectra of α-glucosidase (0.35 µM) and inhibitor (0.05–1.5 µM) were carried out at three different temperatures (298, 304 and 310 K), and the bandwidths were set at 5 nm for both emission and excitation slits. For each sample, three fluorescence spectra were acquired, and the blank was subtracted. The compound quenching mechanism was evaluated using the Stern–Volmer equation [51]. Synchronous fluorescence spectra were collected in the emission range of 260–320 nm [52]. The difference between excitation and emission wavelength (Δλ) was established at 15 nm (for tyrosine residues) or 60 nm (for tryptophan residues) [53].

#### 3.5.2. Circular Dichroism Measurements

To investigate changes in the secondary structure of α-glucosidase in the presence of α-mangostin, circular dichroism (CD) spectra were collected using the Jasco J-810 circular dichroism spectropolarimeter (Tokyo, Japan) at wavelengths between 200 and 250 nm in a nitrogen environment (1 atm). All solutions were prepared in PBS (pH 6.8). The quartz cuvette used had a path length of 1 cm. The concentration of α-glucosidase was 1 µM. The molar ratios of the enzyme/inhibitor were 1:0, 1:1, and 1:2. The blank and PBS signals were removed to produce an accurate background signal. Quantification of the different components of the secondary structure of the enzyme was established using the online SELCON3 program (accessed on 20/12/2021) [54].

### 3.6. Statistical Analysis

Data are expressed as mean ± SEM of at least three independent experiments. Results were analyzed using Microsoft Excel for Windows 10, while sigmoid curve fitting and statistical evaluations were performed using GraphPad Prism 6 (San Diego, CA, USA). The half maximal inhibitory concentration (IC_50_) was estimated by nonlinear regression. The difference between controls and each treatment was assessed using Student’s *t* test. Statistical comparisons among three or more groups were performed using ANOVA, followed by Tukey’s multiple comparison test. The level of significance was established at *p* < 0.05.

## 4. Conclusions

α-Mangostin exhibits antiglycation, antioxidant, and anti-α-glucosidase activities, while synthetic compounds CS1–CS4 showed antioxidant activity and inhibited albumin glycation but missed the inhibition of the enzyme. Thus, the xanthenone nucleus requires further structural functionalization toward the development of improved antidiabetic agents with easily accessible chemical synthesis. In detail, α-mangostin decreased the enzymatic activity in a concentration-dependent manner in the micromolar range by a reversible mixed-type inhibition by changing the secondary structure in the tyrosine microenvironment, increasing the α-helix and random coils.

This research provides new information on the inhibitory mechanism of α-mangostin on α-glucosidase; however, additional in vitro and in vivo studies are required for candidate α-mangostin in the therapy of diabetes mellitus.

## Data Availability

Not applicable.

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
