# Peer review of "Anti-α-Glucosidase and Antiglycation Activities of α-Mangostin and New Xanthenone Derivatives: Enzymatic Kinetics and Mechanistic Insights through In Vitro Studies"

_molecules, 2022, doi:10.3390/molecules27020547_

Round 1

Reviewer 1 Report

In the current manuscript, the authors evaluated antioxidative properties, activity against protein glycation and potential for inhibition of α-glucosidase of natural compound α-mangostin, as well as of four synthetic derivatives of xanthenone. They presented very well the mechanism of α-glucosidase inhibition by α-mangostin. The manuscript is written in a clear and concise manner, and the results highlighted potential role of α-mangostin in preventing complications of diabetes mellitus. Overall, the data presented in the manuscript support the need of further studies in order to reach a firm conclusion on the role of these potential antidiabetic agents.

Comments

  1. Can you briefly discuss the mechanism responsible for anti-glycation activity (anti-AGE activity) of these compounds?
  2. Could you possible briefly mention what is known about toxicity of xanthones?
  3. Please include information on number of repetitive experiments for the data presented in Figures 1 and 2 and Table 1.

Reviewer 2 Report

In this paper, the anti-α-glucosidase and anti-glycosylation activities of α-mannoside and four new derivatives (CS1-CS4) were studied. The results showed that α-mannoside and four new derivatives (CS1-CS4) had anti-glycosylation activity, and only α-mannoside had anti-α-glucosidase activity. From the results, only CS2 of four new derivatives had the same anti-glycosylation activity as α-mannoside. And anti-free radical activity of α-mannoside was stronger than that of four new derivatives. Synthesis of new derivatives, in the design, is certainly expected to be better than natural α-mannoside in efficacy. The efficacy of new derivatives (CS1-CS4) in this paper is not as good as that of natural α-mannoside. So what is the significance of this experiment? In the introduction, the author mentioned that “molecular model studies have shown that flavanone derivatives can bind to the ectopic sites of α-glucosidase” and “further improvement of inhibitory activity can be observed, and the combination of oxygen or nitrogen-containing groups on these aromatic units”. Do the experimental results contradict them? Therefore, α-mannoside must be emphasized in the topic.

Round 2

Reviewer 2 Report

The author added the content In the "Discussion", which explained the meaning of the article and answered the questions by reveiw. It provides the experimental basis for the structural modification of natural compound α-mannoside.